# Characterization of the RNA Transcription Profile of *Bombyx mori* Bidensovirus

**DOI:** 10.3390/v11040325

**Published:** 2019-04-03

**Authors:** Rui Li, Pengfei Chang, Peng Lü, Zhaoyang Hu, Keping Chen, Qin Yao, Qian Yu

**Affiliations:** Institute of Life Sciences, Jiangsu University, Zhenjiang 212013, China; 18086525387@163.com (R.L.); 15751000531@163.com (P.C.); penglu@ujs.edu.cn (P.L.); 15189184890@163.com (Z.H.); kpchen@ujs.edu.cn (K.C.)

**Keywords:** *Bombyx mori* bidensovirus, RACE, RT-qPCR, transcription mapping, overlapping promoters

## Abstract

*Bombyx mori* bidensovirus (BmBDV) is a single-stranded DNA (ssDNA) virus from the genus *Bidensovirus* of the Bidnaviridae family, which, thus far, solely infects insects. It has a unique genome that contains bipartite DNA molecules (VD1 and VD2). In this study, we explored the detailed transcription mapping of the complete BmBDV genome (VD1 and VD2) by rapid amplification of cDNA ends (RACE), reverse transcription quantitative real-time PCR (RT-qPCR), and luciferase assays. For the first time, we report the transcription map of VD2. Our mapping of the transcriptional start sites reveals that the NS genes in VD1 have separate transcripts that are derived from overlapping promoters, P5 and P5.5. Thus, our study provides a strategy for alternative promoter usage in the expression of BmBDV genes.

## 1. Introduction

*Bombyx mori* bidensovirus (BmBDV) is a unique bipartite DNA virus that currently represents the only species in the genus *Bidensovirus* of the Bidnaviridae family [1]. BmBDV exclusively infects the columnar cells of the larvae midgut epithelium, causing chronic densonucleosis disease. The virion of BmBDV has a non-enveloped, spherical, icosahedral structure, 20–24 nm in diameter, and packages a linear single-stranded DNA molecule from the VD1 (~6.5 kb, GenBank accession no. NC_020928) or VD2 (~6 kb, GenBank accession no. NC_020927) genome [2,3]. Furthermore, both VD1 and VD2 genomes are characterized by having inverted terminal repeats (ITRs) at their ends that form a panhandle structure and share a common terminal sequence (CTS) of 53 nts [4]. VD1 contains four open reading frames (ORFs; ORF1 to ORF4), which encode nonstructural protein 2 (NS2) [5], nonstructural protein 1 (NS1) [6], a major structural protein (VP) [7,8], and DNA polymerase (PolB) [9], respectively. VD2 contains two ORFs, encoding a nonstructural protein 3 (NS3) [3] and a minor capsid structural protein (P133) [10]. NS1 is a multifunctional protein, which is similar to the NS1 protein in parvoviruses, likely possesses activities involving ATPases, site-specific DNA binding, endonucleases, and helicases [6,11]. Thus, it is essential for various processes associated with virus replication [12]. NS2 shares no homology with the NS2 found in protoparvoviruses [5]. However, it may be an integral membrane protein, like the adenovirus death protein (ADP) [13], that may promote cell lysis and virus release [5]. The exact function of the NS2 protein is unknown [14]. PolB is homologous to family B DNA polymerase, which is involved in protein-primed replication [15,16]. BmBDV is the only virus that possesses an ssDNA genome and encodes a DNA polymerase [9,17]. The function of NS3 is unknown. Previous studies have shown that NS3 shares homology with the NS3 of *Galleria mellonella* densovirus (GmDV) [18] and the ORF11 in *Plodia interpunctella* granulosis virus (PiGV) [19], which may play an important role in virus replication [20]. P133 is the largest viral structural protein of BmBDV, and it is similar to the VP3 protein of the *Bombyx mori* cytoplasmic polyhedrosis virus (BmCPV), which belongs to the *Reoviridae* family [10,21]. The amino acid sequences of P133 in the leucine zipper region were conserved and protein homologues occur mainly in the outer layer of the viral capsid, so P133 may interact with viral DNA and be related to virus invasion of host [22]. In addition, BmBDV replicates its genome using a unique DNA replication mechanism that does not follow the typical rolling circle replication model initiated by an enzyme replicator, as other known ssDNA viruses do. Thus, the strategy of BmBDV replication is of great interest to investigate.

Previously, the transcription strategy of the VD1 genome of BmBDV has been studied to some extent. Transcript mapping [23] shows that VD1 produces three mRNAs of 1.1 kb, 1.5 kb, and 3.3 kb in size, respectively. The nonstructural proteins (NS1 and NS2) are expressed by alternative initiation codons from a 1.1 kb mRNA transcript [1], and the major structural proteins (VPs) are expressed by a leaky scanning mechanism [7,8] from a 1.5 kb mRNA transcript. The 3.3 kb mRNA transcript contains the ORF4 that encodes DNA polymerase (PolB). Alternative splicing of mRNA transcripts was not observed during BmBDV gene expression and the transcription strategy of VD2 has not been studied.

In this report, we explore the complete transcription strategy of both the VD1 and VD2 genomes. We find that VD1-NS gene has two separate transcripts, which may be controlled by two overlapping promoters (P5/5.5), which differ from previous studies. We present an analysis of the transcription modalities of NS in VD1. We analyze the NS transcripts of VD1 by reverse transcription quantitative real-time polymerase chain reaction (RT-qPCR) as well as the activity of overlapping promoters’ (P5/5.5) activities using a dual-luciferase reporter assay system.

## 2. Materials and Methods

### 2.1. Insect Rearing and Virus Propagation

A major impediment to bidensovirus studies is the lack of permissive insect cells that support virus replication in vitro [24]. Therefore, we used a variant of *Bombyx mori* silkworm (Jingsong × Haoyue), which is sensitive to BmBDV infection. The preserved silkworm eggs stored at 4 °C were removed from the refrigerator for pickling to hatch the diapause silkworm eggs. After the larvae had hatched from the eggs, we conducted timely feedings of silkworm larvae using moderately clean and fresh mulberry leaves. On the first day of the fifth instar, the BmBDV solution (5 μL per head) was freshly prepared and used to feed silkworms via oral instillation. We obtained the midguts of silkworms from BmBDV-infected larvae at 24, 48, 72, and 96 h post-infection (hpi), respectively, by dissecting silkworms at different phases. Silkworm midguts were kept in RNAlater^®^ Solution (Invitrogen, Carlsbad, CA, USA) at −80 °C prior to RNA preparation.

### 2.2. Viral mRNA Extraction and RT-PCR

Total RNA was isolated from BmBDV-infected silkworm midguts (24, 48, 72, and 96 hpi) using Trizol^®^ Reagent (Invitrogen, Carlsbad, CA, USA), as previously reported [24]. The quality of RNA was detected by its absorption value at 260 nm and gel electrophoresis on a 1% agarose gel. The mRNA was extracted using the Poly(A)Purist™ MAG Kit (Invitrogen, Carlsbad, CA, USA) according to the manufacturer’s instructions. Reverse transcription (RT)-PCR was performed to identify viral mRNA transcripts from infected larvae and to determine the optimal time to analyze viral mRNA. The primers used are listed in Appendix A.

### 2.3. Identification of the 5′ and 3′ Ends of Viral Transcripts

The full length of cDNA of each gene and the 5′ starts and 3′ ends of the viral transcripts were determined by rapid amplification of cDNA ends (RACE) using SMARTer^®^ RACE 5′/3′ Kit (Clontech, Dalian, China) in accordance to the manufacturer’s instructions. Primers were synthesized at Generay Biotech Co, Ltd. (Shanghai, China). Primer sequences are shown in genes in Appendix A, and BmBDV-specific primers are shown in Appendix A. All PCR products (see Appendix A) used for the RACE experiments were subsequently cloned into pEASY^®^-T3 cloning vectors (Transgen, Beijing, China) and submitted for sequencing (see Appendix A) at Sangon Biotech Co, Ltd. (Shanghai, China).

### 2.4. Analysis of NS Transcripts from VD1 Using RT-qPCR

The NS1 and NS2 gene segments were amplified using RT-PCR, and the PCR products were extracted using the E.Z.N.A.^®^ Cycle-Pure Kit (Omega, GA, USA). Standard curves of NS1 and NS2 mRNAs were drawn (see Appendix A) using the 7300 Fast system (Applied Biosystems, Foster City, CA, USA). The locations of primers targeting NS transcripts are shown in Appendix A, and their sequences are shown in Appendix A. The forward primer F2 was unique to NS2 transcripts. Primer F2/R was used to specifically amplify the NS2 transcript, and primer F1/R was used to amplify both transcripts.

### 2.5. Activities of the Overlapping Promoters P5/5.5

We made two constructs (P1 and P5) (see Appendix A) that have the NS1 and NS2 transcription initiation sites (Inr1 and Inr2) to analyze promoter activities (Figure 4A), in which the intact upstream promoter elements were cloned into the luciferase reporter plasmid (PGL3-basic). Various mutants were generated, in which all ATGs were mutated to ACCs, or the Inr1 (CATT) for the NS1 initiation site was mutated to TTTT, and the TATA1 for NS1 TATA box (TATA1) was mutated to GCGC. Different constructs containing the intact luciferase initiation codon (P1 to P6) served as a positive control for transcription, whereas those lacking the luciferase initiation codon (P1- to P6-) served as reporters. The construct, together with pRL-ie1 was co-transfected into two insect cells (BmN and Hi5) using Cellfectin^®^ reagent (Invitrogen, Carlsbad, CA, USA) in accordance with the manufacturer’s instruction. Cells were harvested at 48 hpi, and luciferase activity was determined using the Dual-Luciferase^®^ Reporter Assay System (Promega, Madison, WI, USA) as previously described in reference [25], using the pGL3 luciferase reporter vectors and the pRL-ie1 vector. All of the constructs in this study were verified by sequencing.

## 3. Results

### 3.1. Mapping of the Transcripts by 5′/3′ RACE

The full-length cDNA of each gene is shown in Figure 1. Each gene corresponding to a full-length cDNA band, except for NS3, is indicated. Two bands were detected in lane 6 (NS3). The lower band was the same size as predicted, and the upper band was verified to be a non-specific amplified band.

The results of the RACE experiments showed that each transcript corresponded to an ORF, thus alternative splicing of any mRNA transcripts was not observed. For VD1, in contrast to what was reported previously from one transcript, it was clear that NS1 and NS2 were transcribed from two different transcripts (Figure 2A) [23]. The NS2 transcript started at nt 290, and the NS1 transcript started at nt 316, 3 nts downstream of the NS2 initiation codon. NS2 started at nt 290, which is located at the TATA box of the promoter transcribing NS1. All NS1 and NS2 transcripts terminated at nt 1438 from one canonical AAUAAA polyadenylation signal. The VP transcript started at nt 1350, upstream of the polyadenylation site for NS transcripts, and terminated at nt 2929, which overlapped with the 3′ end of the NS transcript by 89 nts. Notably, the PolB open reading frame (ORF) initiated at 6 nt downstream of the previously presumed ATG, which was 21 nts downstream of the TATA box of the P97 promoter. Consequently, the complementary strand of the PolB transcript overlapped with the VP transcript by 4 nt, which warrants further investigation for any regulatory function.

For VD2, NS3 started at nt 565, 19 nts downstream of the P10 promoter TATA box at the 5′ end of VD2. The P133 transcript started 24 nt downstream of the P89 promoter TATA box at the 3′ end of VD2 (nt 5367), only 10 nts upstream of the first ATG, and it terminated at nt 1771. In contrast to VD1, NS3 and P133 transcripts terminated at two polyadenylation sites, which were far apart from each other, unlike like VP and PolB mRNAs, which had 4 nts of overlap.

### 3.2. Analysis of NS Transcripts in VD1

Using 5′ RACE, we observed that two NS transcripts were initiated from different locations on VD1 (Figure 3). This is in contrast to a previous study that showed the NS1 and NS2 transcripts shared the same start site from nt 316 [23]. In order to verify the transcription of NS1 and NS2, we used RT-qPCR to determine the copy numbers of NS1 and NS2 transcripts. Specific primers of the NS2 transcript (F2/R1-2) (Appendix A) were used to amplify NS2-specfic transcripts. Surprisingly, specific primers of NS (F1/R) (Appendix A) amplified not only NS1 transcripts but also NS2 transcripts. The NS2 transcription of NS2 was later than that of NS1 because the NS2 transcripts were not observed at 24hpi (Figure 4), but both NS2 and NS1 transcripts were detected from 48 to 96 hpi. Therefore, we conclude that NS1 and NS2 are not transcribed from the same promoter.

### 3.3. The Function of Overlapping Promoters (P5/5.5)

The results of RACE and RT-qPCR revealed that the NS genes in VD1 expressed separate transcripts. Since there were two TATA box elements upstream of the transcription initiation of NS1, we speculated that the NS1 transcripts may be under the control of two overlapping promoters (P5/5.5). To test this hypothesis, various reporter plasmids were constructed (Figure 4A), and were co-transfected with pRL-ie1 into insect cells (BmN, Hi5), followed by analysis of the function of the two overlapping promoters using luciferase assays. We constructed plasmids (P1 to P4) (Figure 4A) to clearly understand the functions of the two overlapping promoters (P5/5.5). The plasmids (P1 to P4) were expected to express both NS1 and NS2 mRNA. However, we observed low luciferase activity of P1 and P2 (Figure 4B), which came from the NS2-fused protein with luciferase, as direct translation or leaky scanning from the luciferase initiation codon depends on NS1 or NS2 transcripts. Unexpectedly, P1- had no activity when the luciferase initiation codon (ATG) was mutated. This indicated that P5/5.5 had weak activity, which may have been insufficient to start NS2 transcription. In addition, the NS2 initiation codon may not be functional. Therefore, we constructed P3 and P4 by mutating the NS2 initiation codon (ATG) for further examination. Obviously, the results indicated that the activity of P3 and P4 increased, and P3 yielded a 1.3-fold increase and P4 yielded a 2.6-fold increase in luciferase activity, respectively, compared to P1 and P2, in Hi5 cells. Hence, we concluded that the NS2 initiation codon affects the translation of luciferase.

To investigate the function of P5/5.5 in NS1 transcription, we constructed plasmids (P5 to P8) (Figure 4A). We already knew that the NS2 initiation codon negatively regulates luciferase expression. As expected, P5 induced a level of activity that was low but higher than that of P1, suggesting that the promoter may be under the control of a downstream promoter element (DPE). Previous studies have suggested that a DPE consensus, RGWYV(T), is located about 28 to 33 nts downstream of the transcription start site [26]. Particularly, we found a DPE-like sequence, GGTCA, located at 29 nts to 33 nts downstream of Inr1. In order to explore impacts on transcription of NS1 and NS2, the P6**^Δ^** observed through the DPE-like sequence GGTCA in P6 was mutated to CCGA. As expected, the luciferase activity of P6**^Δ^** declined rapidly compared to that of P6.

Transcription initiation is one of the most important control points in the regulation of gene expression [27]. P2***** and P6***-** only generated NS2 transcripts of NS2 by knocking out TATA1 and Inr1. As indicated from the results, the activity of P2***** was similar to that of P2. These results indicate that the two overlapping promoters (P5/5.5) had weak activity levels, which further proved our previous hypothesis. P6***-** and P6- had high levels of activity, and the initiation of NS1 and NS2 transcription was closely positioned at either side of the NS2 initiation codon (Figure 2A). Consequently, NS1 could be generated from the NS2 transcript by alternative initiation. In order to investigate the function of NS1 promoter elements for NS2, we made P6****-** by knocking out TATA2 in P6***-**. The results showed that the activity declined drastically in luciferase activity compared to that of P6***-**. However, P6****-** also presented a lower level of activity. In conclusion, our deletion analyses suggest that the promoter can induce TATA-box-independent transcription.

## 4. Discussion

BmBDV was classified as a member of the family *Bidnaviridae* by the International Committee on Taxonomy of Viruses (ICTV) in 2012 [1]. At that time, it was the only species of the *Bidensovirus* genus identified in insects. With the development of metagenomics, more and more viruses have been found in environmental and biological samples [28]. Recently, the presence of ssDNA viruses of the *Bidnaviridae* family in sponges and corals has been reported [29]. 

In this study, we analyzed the transcription strategies of BmBDV. Transcription mapping was used to show the transcription initiation and termination sites of each gene in BmBDV (see Figure 2). Interestingly, Figure 2A shows that the transcription initiation sites of NS1 and NS2 are closely positioned at either side of the NS2 initiation codon, so NS1 can be generated from the NS2 transcript by alternative initiation. This characteristic is similar to that of *Aedes albopictus* densovirus (AalDV) [30] during transcription. NS1 and NS2 may be encoded from separate transcripts (see Figure 3), which are transcribed by two overlapping promoters. In fact, as shown in Figure 4A, the promoter driving NS2 transcription is the P5 promoter, which contains TATA2 and Inr2. However, since there are two TATA box elements upstream of the transcription initiation of NS1, NS1 transcription may be under the control of P5/5.5. We also found that in VD1, P5/5.5, which drives the transcription of the NS gene is a weak promoter, but its activity is enhanced in the presence of DPE. In BmBDV, both the TATA box and Ins play important roles in the high activity of the promoter driving NS1 and NS2, but are not sufficient to drive the transcription of NS1 genes in BmBDV [25]. Therefore, the transcription of NS2 and NS1 are synergistically driven by overlapping promoters and DPE. Therefore, the expression of NS1 is guaranteed by alternative and overlapping promoters. We know that NS1 is a multifunctional protein that is essential for various processes involved in viral propagation [12]. Obviously, the transcription profile of VD1-NS genes benefits the expression of NS1. Based on the phylogenetic tree analysis of parvovirus NS1, it has been speculated that the *Brevidensovirus* genus is the ancestor of other parvoviruses [31]. AalDV belongs to the *Brevidensovirus* genus and perhaps employs the above transcription strategies to ensure the stable inheritance of AalDV.

Remarkably, despite our finding that NS2 ATG negatively regulates the activity of P5/5.5, P1 has no activity (see Figure 3B) when the NS2 transcription is merely driven by P5 (see Figure 3A); the NS2 initiation codon does not exhibit activity, since P5 is insufficient to initiate NS2 transcription in the absence of DPE. Unexpectedly, the activity of P5/5.5 dramatically increased after the NS2 ATG was mutated (see P3, P4, and P6 in Figure 3B). However, the function of the NS2 initiation codon for P5/5.5 promoters cannot be completely understood using current studies. We hypothesize that NS2 ATG may be in a region that is negatively regulated for the P5/5.5 promoters. The mutation of NS2 ATG can destroy the sequence completion in the original region, causing its regulatory function to cease. We are equally uncertain about whether NS2 ATG has dual functions. This assumption has not been reported before and may be a new finding. Therefore, we need to further investigate into the function of the nearby area of the NS2 ATG.

BmBDV exclusively infects the columnar cells of the larvae midgut epithelium. A major impediment to *Bidensovirus* studies is the lack of permissive insect cells that support virus replication in vitro [32]. BmBDV can be rescued from BmN cells transfected with recombinant plasmids containing the linear full length virus genome [24]. BmBDV has the potential to be a viral vector [33]. To achieve controllable transformation of the viral genome, it is necessary to clarify the transcription strategies of the viral genes. In this study, for the first time, we reported the transcription strategy of VD2 (see Figure 2B). Interestingly, we showed, through mapping, that NS3 and P133 transcripts are terminated at two polyadenylations sites, which are far apart from each other. Therefore, this non-transcribed region could be a suitable locus for the insertion of exogenous genes in a future Bidensoviral vector. In conclusion, our study provides new information that could be used to employ BmBDV as viral vectors and biological control tools in the future.

## Figures and Tables

**Figure 1 viruses-11-00325-f001:**
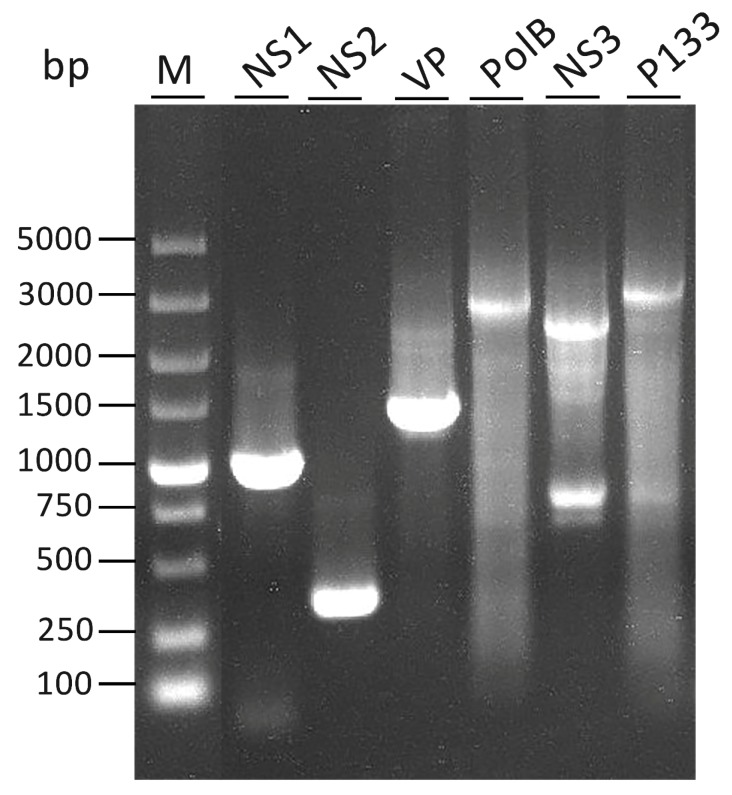
Detection of the full length of cDNA of each gene. Two bands appear in lane 6 (NS3). The lower band was the same size as the predicted, and the upper band has been verified as a non-specific band. The 5’RACE-Ready cDNA was amplified using reverse gene-specific primers (GSPs) (NS1endR, NS2endR, VPendR, PolBendR, NS3endR, P133endR) in Appendix A, and universal primer mix (UPM) in the kit.

**Figure 2 viruses-11-00325-f002:**
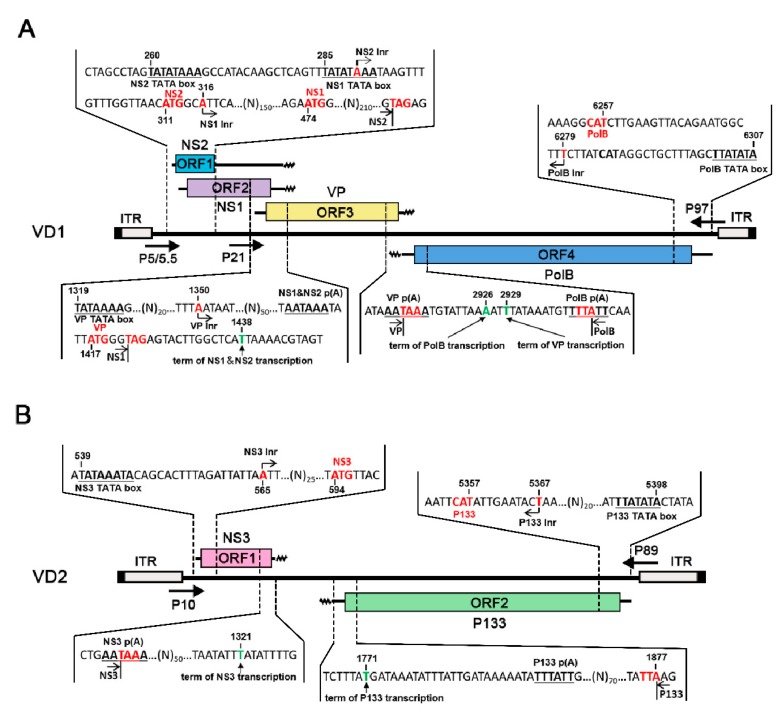
Mapping of 5′ and 3′ ends of *Bombyx mori* bidensovirus (BmBDV) transcripts: (**A**) mapping of 5′ and 3′ ends of VD1, (**B**) mapping of 5′ and 3′ ends of VD2. The bars with inverted terminal repeats (ITRs) at both ends represent the BmBDV genome. Major open reading frames (ORFs) that encode proteins are shown as boxes, whereas mRNA transcripts are indicated in the middle. The wavy lines indicate polyA tails. The arrows indicate the position of the VD1 and VD2 promoters.

**Figure 3 viruses-11-00325-f003:**
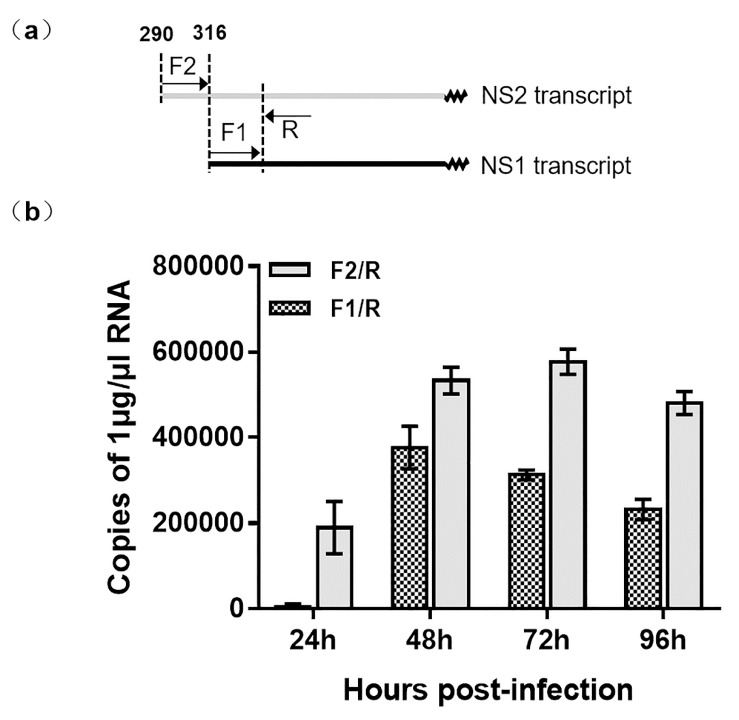
Analysis of NS transcripts of VD1 by RT-qPCR. (**a**) Locations of primers for RT-qPCR. The forward primer F2 is a specific primer for the NS2 transcript, and F1 is shared by the NS2 and NS1 transcripts, which have the same reverse primer, R. (**b**) The copy numbers of the products are amplified. The gray column represents the copy numbers of mRNA products amplified by primer F2/R, and the shadow column indicates the copy numbers of the products amplified by primer F1/R. (n=3), Error bars denote standard deviation.

**Figure 4 viruses-11-00325-f004:**
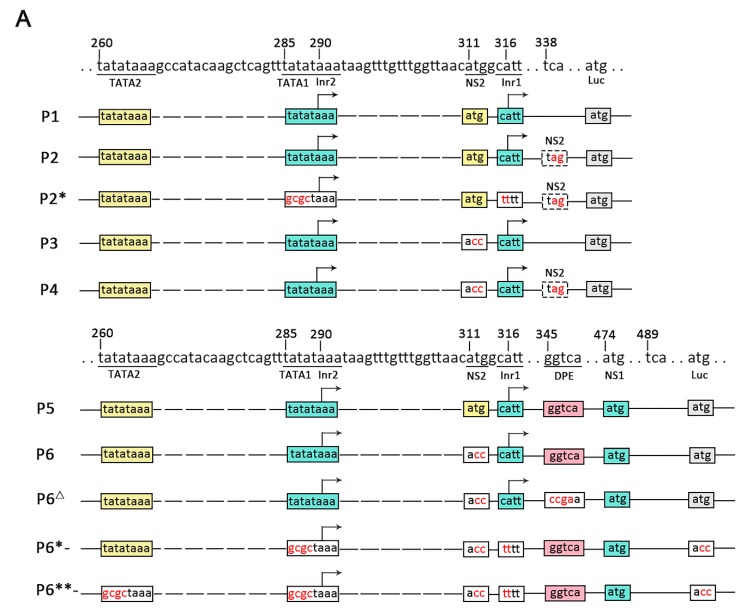
Analysis of P5/5.5 promoter elements using the luciferase reporter system. (**A**) The luciferase reporter plasmids contain NS1 and NS2 transcription initiation sites (Inr1 and Inr2), an NS1 or NS2 initiation codon (ATG), and its intact upstream promoter elements. In the constructs, the boxes represent elements, and the gray boxes represent replaced sequences. In addition, mutants were made for all constructs in which the initiation codon of luciferase was mutated (P1- to P6-), and two constructs, TATA1 and Inr1 (for NS1 transcripts), were mutated (indicated by *****). ****** represents Inr1. TATA1 and TATA2 all were mutated, and then DPEs were mutated (indicated by **^Δ^**). (**B**) The observed luciferase activity. The tendency of activity levels to change in different insect cells (Hi5, BmN) was consistent (“l.s.” represents luciferase activity if leaky scanning occurs and “e.p.” represents overall expected activities). (n = 3), Error bars denote standard deviation.

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
