# Peer review of "Characterization of the RNA Transcription Profile of Bombyx mori Bidensovirus"

_viruses, 2019, doi:10.3390/v11040325_

Reviewer 1 Report

The term "Transcription Strategy" suggests that there is some strategy -the relevant Mirriam Webster definition of “strategy” would be "an adaptation or complex of adaptations (as of behavior, metabolism, or structure) that serves or appears to serve an important function in achieving evolutionary success". However, the manuscript shows nothing of the kind, but does show detailed transcript mapping. Therefore, please change "transcription strategy" to "detailed transcript mapping" or something similar throughout the manuscript, and correct sentences where appropriate so they make sense. Line 169: “To  prove  this …” In science, we gather data to support a hypothesis, we don’t prove things. Change to “To test this hypothesis, …) Line 186: “As expected, P5 executed level of activity …” change to “As expected, P5 induced a level of activity …” Line 213: “ … can exhibit TATA-less transcriptional activities.” change to “can induce TATA-box-independent transcription.” Line 219: “We believe that insects are not the only hosts of Bidensovirus species”. This is pure speculation without any supporting evidence or citations. Please delete this statement. Line 220: “The comprehensive analysis of  transcription  strategies of BmBDV presented in this study provides guidance for the discovery of novel  members of the Bidnaviridae family in future research. “ I don’t see how transcript mapping will aid in the future discovery of related viruses. Please delete this statement. Line 263: “This region of non-transcription is a beneficial place for a vector to insert exogenous genes  without affecting the expression and regulation of the virus”. This is purely speculative – there is no information on this as no Bidensoviral vectors have been developed. Please change to “Therefore, this non-transcribed region could be a suitable locus for the insertion of exogenous genes in a future Bidensoviral vector.” Line 264: The last sentence doesn’t make sense. Please change to “In conclusion, our study provides new information that could be used to employ BmBDV as viral vectors and biological control tools in the future.  “

Author Response

Point 1: The term "Transcription Strategy" suggests that there is some strategy -the relevant Mirriam Webster definition of “strategy “would be "an adaptation or complex of adaptations (as of behavior, metabolism, or structure) that serves or appears to serve an important function in achieving evolutionary success". However, the manuscript shows nothing of the kind, but does show detailed transcript mapping. Therefore, please change "transcription strategy" to "detailed transcript mapping" or something similar throughout the manuscript, and correct sentences where appropriate so they make sense.

Response 1: Considering the Reviewer’s suggestion,we would like to change our title as :“Characterization of the RNA transcription profile of Bombyx mori bidensovirus”

Point 2: Line 169: “To prove this …” In science, we gather data to support a hypothesis, we don’t prove things. Change to “To test this hypothesis, …)

Response 2: We have made correction according to the Reviewers comments.

Point 3: Line 186: “As expected, P5 executed level of activity …” change to “As expected, P5 induced a level of activity …”

Response 3: We have made correction according to the Reviewer’s comments.

Point 4: Line 213: “… can exhibit TATA-less transcriptional activities.” change to “can induce TATA-box-independent transcription.”

Response 4: We have made correction according to the Reviewer’s comments.

Point 5: Line 219: “We believe that insects are not the only hosts of Bidensovirus species”. This is pure speculation without any supporting evidence or citations. Please delete this statement.

Response 5: We have deleted We believe that insects are not the only hosts of Bidensovirus species according to the Reviewers comments.

Point 6: Line 220: “The comprehensive analysis of transcription strategies of BmBDV presented in this study provides guidance for the discovery of novel members of the Bidnaviridae family in future research. “ I don’t see how transcript mapping will aid in the future discovery of related viruses. Please delete this statement.

Response 6: We have deleted the statement according to the Reviewers comments.

Point 7: Line 263: “This region of non-transcription is a beneficial place for a vector to insert exogenous genes without affecting the expression and regulation of the virus”. This is purely speculative – there is no information on this as no Bidensoviral vectors have been developed. Please change to “Therefore, this non-transcribed region could be a suitable locus for the insertion of exogenous genes in a future Bidensoviral vector.”

Response 7: We have changed the sentence according to the Reviewers comments.

Point 8:Line 264: The last sentence doesn’t make sense. Please change to “In conclusion, our study provides new information that could be used to employ BmBDV as viral vectors and biological control tools in the future. ”

Response 8:   We have changed the sentence according to the Reviewers comments.

Reviewer 2 Report

This study provides a detailed transcriptional analysis of BmBDV VD1 and VD2 single-stranded DNA.  Whilst DV1 has been partially characterised before, this is the first report for VD2.  The authors obtain some different results to that previously published and therefore it is important that the data are as robust as possible.  Therefore, in Figure 3 and 4 it should be stated how many times the expt was repeated (n=?) and what the error bars stand for.  For Figure 3 B, are the comparisons noted statistically significant?  What stat tests were done?

The data are generally clearly laid out and the conclusions drawn are supported by the data.  I would just like to see that the data are robust.

Other points:

Introduction line 43 - what role, explain further

Results line 118/19 - How was non-specificity verified?

             line 151 - ....previous study that SHOWED the ....

             line 152 -  transcription

             line 147 - polyadenylation; clarify what is meant by 'far apart from each other'

Figure 3b - vertical axis title 'copies'

Figure 3 legend line 157 - 'reward' do you mean 'reverse' ?

Figure 2 line 143 - wavy; position

Author Response

Point 1: This study provides a detailed transcriptional analysis of BmBDV VD1 and VD2 single-stranded DNA.  Whilst DV1 has been partially characterised before, this is the first report for VD2.  The authors obtain some different results to that previously published and therefore it is important that the data are as robust as possible.  Therefore, in Figure 3 and 4 it should be stated how many times the expt was repeated (n=?) and what the error bars stand for.  For Figure 3 B, are the comparisons noted statistically significant?  What stat tests were done? The data are generally clearly laid out and the conclusions drawn are supported by the data.  I would just like to see that the data are robust.

Response 1: To determine accuracy of experimental results,  in this study,  it is necessary to repeat the experiment many times. In figure 3, n=3,  the error bars stand for standard deviation. In figure 4B, n=3, the error bars stand for standard deviation.

In figure 4B, each set of data is representative of three independent experiments, each performed in triplicate. The activity of each construct is expressed as a ratio of that to the promoterless control plasmid pGL3-Basic (The values is 1). The comparisons display the tendency of activity levels to change, have no statistically significant.

Point 2: Introduction line 43 - what role, explain further

Response 2: We are very sorry for our negligence of the role of P133. At present, the function of P133 is unknown. However,the amino acid sequences of P133 in the leucine zipper region were conserved and protein homologues occur mainly in the outer layer of the viral capsid, so P133 may interact with viral DNA and be related to virus invasion of host[23]. (Weve added this part in the revised version)

23. Cotmore, S. F.; Tattersall, P., Parvoviral host range and cell entry mechanisms. Advances in virus research 2007, 70, 183-232.

Point 3: line 118/19 - How was non-specificity verified?

The upper band (in figure 1, lane 6) was extracted and cloned into pEASY®-T3 cloning vectors. The sequencing analysis indicated the upper product (upper band) is not specific to NS3 related amplicon.

Point 4:  line 151 - ....previous study that SHOWED the ....

Response 4: We have made correction according to the Reviewers comments.

Point 5:  line 152 -  transcription

Response 5: We have made correction according to the Reviewers comments.

Point 6:  line 147 - polyadenylation; clarify what is meant by 'far apart from each other'

Response 6: We have made correction according to the Reviewers comments.

 NS3, and P133 transcripts terminated at two polyadenylation sites, (NS3 transcript is terminated at nt 1321, P133 transcript is terminated at nt 1771) and both transcripts have no overlapping sequence region. Weve changed the 'far apart from each other' to have no overlapped region.

Point 7: Figure 3b - vertical axis title 'copies'

Response 7: We have made correction according to the Reviewers comments.

Point 8: Figure 3 legend line 157 - 'reward' do you mean 'reverse' ?

Response 8: We mean 'reverse'. We have made correction according to the Reviewers comments.

Point 9: Figure 2 line 143 - wavy; position

Response 9: We have made correction according to the Reviewers comments.
